# Generating Digital Twins for Path-Planning of Autonomous Robots and Drones Using Constrained Homotopic Shrinking for 2D and 3D Environment Modeling

Martin Denk *, Sebastian Bickel, Patrick Steck, Stefan Götz, Harald Völkl and Sandro Wartzack

Engineering Design, Friedrich-Alexander-Universität Erlangen-Nürnberg, 91058 Erlangen, Germany
* Correspondence: denk@mfk.fau.de

**Abstract:** A digital twin describes the virtual representation of a real process. This twin is constantly updated with real data and can thus control and adapt the real model. Designing suitable digital twins for path planning of autonomous robots or drones is often challenging due to the large number of different dynamic environments and multi-task and agent systems. However, common path algorithms are often limited to two tasks and to finding shortest paths. In real applications, not only a short path but also the width of the passage with a path as centered as possible are crucial, since robotic systems are not ideal and require recalibration frequently. In this work, so-called homotopic shrinking is used to generate the digital twin, which can be used to extract all possible path proposals including their passage widths for 2D and 3D environments and multiple tasks and robots. The erosion of the environment is controlled by constraints such that the task stations, the robot or drone positions, and the topology of the environment are considered. Such a deterministic path algorithm can flexibly respond to changing environmental conditions and consider multiple tasks simultaneously for path generation. A distinctive feature of these paths is the central orientation to the non-passable areas, which can have significant benefits for worker and patient safety. The method is tested on 2D and 3D maps with different tasks, obstacles, and multiple robots. For example, the robust generation of the digital twin for a maze and also the dynamic adaptation in case of sudden changes in the environment is covered. This variety of use cases and the comparison with alternative methods result in significant advantages, such as high robustness, consideration of multiple targets, and high safety distances to obstacles and areas that cannot be traversed. Finally, it was shown that the environment for the digital twin can be reduced to reasonable paths by constrained shrinking, both for real 2D maps and for complex virtual 2D and 3D maps.

**Keywords:** path planning; digital twin; skeletonization; thinning; shrinking; autonomous mobile robots; drones

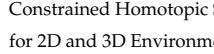



## 1. Introduction

Mobile robots are used today for a variety of different tasks, for example, in the household (vacuum cleaner robots) [1], in logistics [2], factories [3], medicine [4] healthcare [5], and in the sector of security [6]. Especially in the healthcare sector [5], autonomous robots are used in individual work steps to bring or fetch goods or to clean floors. In order to perform the individual tasks, the robots must be equipped with suitable sensor technology to approach the specific paths and to communicate and interact with their environment.

For optimal control and mapping of the environment of a warehouse [2] or a factory [3], a so-called digital twin can be used. Such digital twins usually consist of a digital master enriched with a digital shadow [7,8]. While the master represents the virtual model, the digital shadow serves for the update of e.g., the environment or the robot positions. Based on such an update new maneuver suggestions for each individual robot can be estimated. In this article, the digital master describes the capture of the environment as well as the

individual robots as a digital model similar to [3], which can be updated with a digital shadow by changing robot positions or a changing environment.

### 1.1. Motivation

For global path planning, it is necessary to choose a suitable environment model, an optimization criterion, and a reasonable path finding method [9]. However, since there are a large number of for example logistics and care facilities, such a recognition of the environment must be as completely or partially automated as possible. An initial model in the form of a map can be created by for example building a virtual 3D model [10] scanning the environment with a laser [3] using 2D images [11] or directly predefined maps [12]. Based on this map, the environment has to consider the respective tasks as well as the starting position of the robots and the obstacles. If an environment is available, paths for individual robots can be derived [9]. Based on these paths, process times, clearance widths and environmental influences can be embedded, which can lead to a digital twin to control the individual robots.

In particular, finding possible paths in complex maps to accomplish several tasks is one of the most widespread issues in the robotic domain. In principle, both local and global or coupled local and global path planning can be used [9]. In global path planning, all conditions are already known in the beginning, while in local path planning, conditions such as the robot's position and the map of the environment have to be created first. For care and logistics robots, it is necessary to develop trajectories that respond quickly to changing environments, which furthermore lead also to expectable behavior of the robots. For example, if a robot travels down the center of a hospital corridor, it may be more easily recognized than if it cuts corners and curves.

This work focuses on pathfinding in a dynamically changing environment reducing the possible paths to a subset of topologically centered paths. Therefore, a method for the automated reduction in the 2D and 3D environment to 1D curves is proposed. Therefore, a 2D or 3D map represented as an image is considered, which contains the position of the tasks and the initial robot positions. By using the skeletonization method from [13], such a map is reduced to the 1D curves, which serves as the main data structure (graph) for the digital twin. This approach is applied in this work on virtually generated 2D maps, 3D maps and 2D real world maps generated by a vacuum cleaner (I-Robot). In the following section, a overview of several common path algorithms and concepts are discussed. Furthermore a more detailed overview on applying skeletonization for the reduction in the environment is given, followed by the generation of digital twins in several environments using existing paths.

### 1.2. State of the Art in Path Generation with Skeletonization

If an environment is available, a path to minimize travel time can be found by a common path algorithm based on cost functions such as A*- [14] or Dijkstra-algorithm [15]. Furthermore genetic-based [1], bio-inspired [16,17], or more recently machine learning-based path-finding methods [4,18,19] can also be found. For a more detailed overview in the domain of path-finding, we refer to the review articles [9,20]. In the following, the A*-algorithm is explained in detail as it constitutes a widely used variant.

The A*-algorithm is a heuristic method to find a path from a given source to a given destination [14]. It uses a cost function that chooses the distance of a point to both the source and the destination as the evaluation criterion. However, if a path with the shortest possible distance is chosen as the evaluation criterion, these paths often cut the corners of obstacles, on which for example the digital twin of [10] may touch its obstacles head-on. In addition to the A*-method, a variety of other methods such as Probabilistic Roadmaps (PRM) [21], Rapid Exploring Random Tree (RRT) [22] or Rapid Exploring Random Tree Connect (RRT-C) [23] and Artificial Potential Field (APF) [24] can be used, on which we refer for more details to [24]. These algorithms are used for comparison in Section 3.8 to the proposed path simplification. Although these paths are often the shortest paths to position tools, it

may touch the non-passable area. Moreover, since in reality component deformations due to centrifugal forces or path deviations due to tolerances occur in the robot kinematics, a conservative path detection with a certain distance to the non-passable areas should be rather preferred. Another major drawback of the A*-algorithm is the lack of consideration of multiple targets and sources. Classically, a single optimal path is computed for each target-source combination, rather than planning a holistic route. A more holistic approach to reduce the environment to a subset of paths can be achieved by so-called skeletonization, where multiple destinations and sources can be implicitly considered [12]. The authors of [9] narrowed down the possible paths by a so-called thinning procedure to obtain the most centered path suggestions on which the shortest path is processed with A*. While A* processed on the whole 2D or 3D image environment leads to a massive amount of path possibilities, the reduction with skeletonization leads to an environment consisting only of a few topologically connected path suggestions für A*. Classical trajectory algorithms can capture the shortest path from a source to a target quite efficiently. For this reason, an approach based on skeletonization is used in this work to constrain the possible paths that can also respond to a dynamically changing environment and tackle multiple targets and sources. Furthermore, it is ensured, that the maximum distance of a path to the non-passable areas is implicit embedded in the environment simplification.

The authors of [12,25–27] used a so-called skeletonization to simplify the path search environment. Skeletonization describes a topic in which centered lines (curve skeletons) of 2D geometric objects or centered surfaces/curves of 3D objects can be determined [28]. Such skeletons can be calculated by using for example Blum's concept of the maximal inscribed balls [29]. The center points of circles in 2D or spheres in 3D that touch the boundary of the geometric object at least at two points are used to build the skeleton. However, since this type of calculation is not practical, a number of other concepts have been developed based on these medial skeletons as an analogy, such as the grassfire analogy, Maxwell sets, or symmetry sets [28]. Such skeletonization can now be computed for a variety of geometric descriptions such as images [13,30], point clouds [31], polygon meshes [32], or algebraic functions [33]. For the theoretical background and alternative concepts in the topic of skeletonization, we refer the reader to the book [33], while in [34] the reader will find a more detailed practical explanation of skeletonization. The review articles [28,35,36] include explanations of more recent skeletonization methods, their main advantages and disadvantages, and the most important criteria for success.

Particular for path environment reduction, there are several concepts based on skeletonization such as Voronoi diagrams [25,26], grass-fire also referred to as wave propagation [37,38], potential fields [39,40], or thinning [12,27]. The authors of [25,26] use Voronoi diagrams in which both the environment and the obstacles are constructed as polygons. However, the use of Voronoi diagrams has some disadvantages. For example, if the polygon is noisy, there are many possible branches [28], resulting in a nonrobust path algorithm highly influenced by such noise. An alternative skeletonization method is called thinning, which can be used in pathfinding [27]. On thinning the boundary of the 2D or 3D image is eroded until only 1D lines for 2D images and 1D lines and 2D surfaces for 3D images results. Thinning is slightly more robust compared to a noisy boundary, but often leads not exactly to centered skeletons [28]. Compared to the cost function-based path algorithms, the skeleton-based algorithms offer the advantage of reducing the possible paths to only centered paths that are maximally far from the boundary. In contrast to the use of A* directly without simplified paths [10], these paths are suitable for robust fault management in the digital twin, since there is a quasi-maximal avoidance range to the boundary. Similarly, topologically alternative routes are chosen, e.g., to avoid an obstacle from either side. In addition to generating paths via thinning, the authors of [11] used thinning for preprocessing, which was then proceeded with A* for the final path selection [11]. While in [10] individual obstacles are touched, in [11] central paths in the traversable region result from the combination of A* with path simplification using skeletonization. Therefore, in this work, a thinning-based path-finding method analogous to [27] is used, whose shortest

central path can subsequently be searched using A* or Dijkstra analogous to [27]. However, since 3D maps also play a role particularly for the flight zones of drones or uneven grounds and bridges for mobile robots, a 3D curve skeleton with multiple targets and starting positions based on a modification of the thinning method of [13] is chosen in contrast to [11,27]. This modification is based on so-called anchor points. Anchor points are points that are accepted as skeletal points and thus not eroded during thinning [41,42]. In our work, these anchor points are placed on task and robot positions to ensure connectivity to the path estimation. Likewise, alternative properties for controlling the robots are integrated into the path estimation, to separate recalibration and global path estimation.

Another alternative for the calculation of these skeletons is the use of the extreme values of the so-called distance transformation [28]. In this case, the individual skeleton branches are recorded on the basis of extreme values. Since this information is used directly in the skeletonization, a maximum passage width along the path can be integrated automatically. Therefore, individual robots can be instructed with different-sized packages to only passable paths. In this work, in addition to path generation, the maximum passage width is also extracted based on skeletonization. Since both passage width and skeletonization are based on a similar theoretical model (Blum's maximally inscribed balls), this leads to a robust algorithm without the necessity to cover special cases.

In addition to the actual pathfinding, it may be necessary to place certain safety zones around the robot to allow sufficient time for correction in case of intrusion, e.g., a person [43]. The authors of [43] introduced a so-called fuzzy logic, where uncertainties about the position of individuals are taken into account. While the distance to obstacles is checked locally in this work, a global distance measurement is carried out in this work. A global measurement offers the advantage that it can be decided before the start of the journey whether the destination can be reached at all with sufficient distance to existing obstacles. Similarly, instead of a fixed safety distance, the maximum possible distance to the obstacle should be implicitly calculated for each path alternative.

Another scenario is the misalignment of the robot. Since along a path the uncertainty of the robot position is constantly increasing in the global model, a constant correction of the robot position is necessary [9]. To guide these robots back to the individual paths, heuristic methods or, as in our case, skeletonization can be used. These skeletonization methods allow finding of suitable centered paths for the recalibration of the robots using the same procedure as in the path simplification step without affecting the original path model.

Furthermore, while the works in [25–27] focus on path generation and the works in [2,10] concentrate on the use of path generation of existing twins, this work is mainly concerned with the connection between path generation and digital twin. For example, if the path changes due to altering environmental conditions, this directly leads to an adjustment in the twin. This flexibility enables the fully automated generation of digital twins based on simple map layouts. All combinations of path algorithms and digital twins are based on the use of pathfinding in the digital twin, which does not change during operation as a virtual model (only the states and values in the twin). In this work, however, the digital twin is generated directly from pathfinding, which consider changes in the environment.

In contrast to the alternative works, a holistic approach to trajectory generation as well as repositioning of the robot is presented here. In the following, a comprehensive investigation of virtual and real 2D environments as well as virtual 3D environments is performed. A crucial factor is the linking of trajectory generation with information such as trajectory width from the environment itself. So the main scientific contribution can be summed up as an concept which covers:

- Implicit path finding and clearance width estimation;
- Implicit retracking of robots;
- Implicit multiple tasks and robots in environmental simplification;
- Deterministic path algorithm in 2D and 3D;
- Local updating property;

- Digital twin with state graphs for passability redundant path options and embedded process time (task and path).

At once in one concept. In the following, a method for constructing the environment based on 2D maps is presented first. Then the modification as well as the incorporation of the markers for the maps are explained. Finally, a digital twin is derived based on the path model. The creation of the digital twins is then tested using different map configurations as well as modifications to the map. It should also be possible to return lost robots directly to the existing paths (back to track). In the following drones and mobile robots are referred to as robots.

## 2. Method and Approach

Essentially, the goal now is to obtain digital twins for route descriptions directly from existing scanned 2D and 3D maps. It is assumed that the environment to be traversed can be scanned in such a way that both the areas to be traversed and the position of the individual tasks are available. Such a map is directly acquired or generated, for example, by vacuum cleaner robots via a so-called simultaneous localization and mapping (SLAM) or the manual scan of a work plan.

The map with different events was encoded in RGB. Thus, the map consists of the obstacles, the individual tasks with different colors as well as the initial position of the individual robots. Based on this map a digital twin is created by considering seven steps. Figure 1 shows the different steps generating a digital twin from the RGB map, which leads to a subset of a labeled graph with edges and nodes.

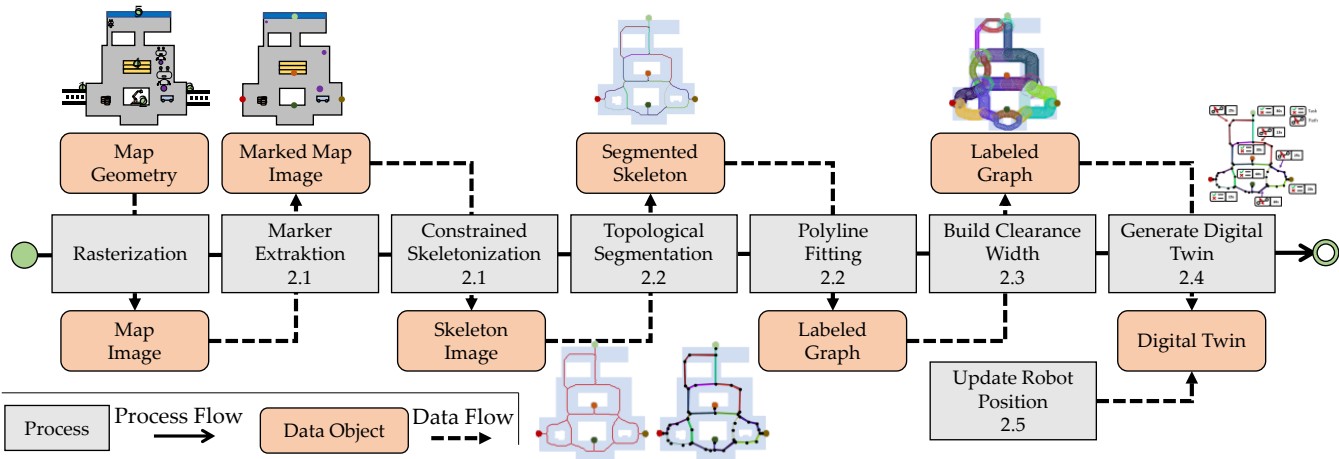

**Figure 1.** Digital twin generation: marker extraction and conditional shrinkage in 2.1, topological segmentation and polyline fitting in 2.2, clearance width estimation in 2.3, digital twin generation in 2.4, and robot position updating in 2.5.

First, the map with the tasks is saved as an RGB image, where the individual tasks, routes and obstacles are coded in the appropriate color. Then the separated events in the image are extracted as markers. A distinction can be made between "passable", "not passable", "position of the robot", and "tasks to be passed" in the marker extraction Section 2.1. Via a shrinkage also referred as constrained skeletonization in Section 2.1 the task markers are used as constrained conditions to calculate the skeleton in the passable area. This skeleton links the individual tasks in the traversable domain. The skeleton can be decomposed into individual path segments using a topological segmentation in 2.2. Since the paths are still in the form of pixel chains, they must be reduced to simple edges and nodes to a graph using a collapsing procedure in 2.2. This graph is then enriched with the local passage width in 2.3. The digital twin can now be obtained directly from the paths as well as from additional information in 2.4. Finally, based on the digital twin, the ability to reposition the robots must be integrated in 2.5, which have left the path with some uncertainty in 2.5. In the following, the individual concepts are explained schematically

using a simple example. For each chapter section, a corresponding result of a more detailed use case can be found in Sections 3.1–3.5. Furthermore, several additional use cases of complex 2D maps in 3.6, of 3D maps in 3.7 and of real world maps in 3.8 are considered.

### 2.1. Constrained Path Estimation with Homotopic Shrinking

The first task is the automatic extraction of the individual markers from the RGB map. Color coding is used to extract the individual tasks from the map. It is important that the individual segments have a unique uniform RGB coding and do not allow any color gradation to the neighboring region. The RGB colors can now be segmented into a passable region, a non-passable region, and tasks. In Figure 2 the passable region is shown in green, purple, and blue as disconnected regions and the non-travelable region is white. Similar coding is also found in the broader map in Section 3.1. The passable and non-passable regions can now be stored in a digital image. A digital image $\wp$ can be stored via a quadruple with

$$\wp = (V, m, n, B) \tag{1}$$

where the two-dimensional elements of $V = \mathbb{Z}^2$ are called pixels of the image [44]. $B$ stands for the pixels in the background and $F = V - B$ for the pixels of the image and therefore of the foreground. The neighborhood can be defined by the configuration $(m, n) = (4, 8)$ or $(m, n) = (8, 4)$. The neighborhood of a pixel $p_i$ with a pixel $p_j$ can be classified by a common edge $n = 4$ or a point contact $n = 8$ [44]. If two pixels share a common edge, they are 4-adjacent to each other. All pixels that are n-adjacent to a point $p_i$ are stored in the set $N_n(p_i)$. A pixel chain $J = \{p_l, \ldots, p_i\}$ with $J \in F$ consists of 8-adjacent points, where the consecutive pixels $p_k$ and $p_{k+1}$ are 8-adjacent to each other [44]. In Figure 2, the green pixel chain represents both a 4-adjacent and an 8-adjacent chain, while the purple pixel chain represents only an 8-adjacent chain. Since the blue region does not allow a chain due to the massive faces, this region must be thinned to a skeleton.

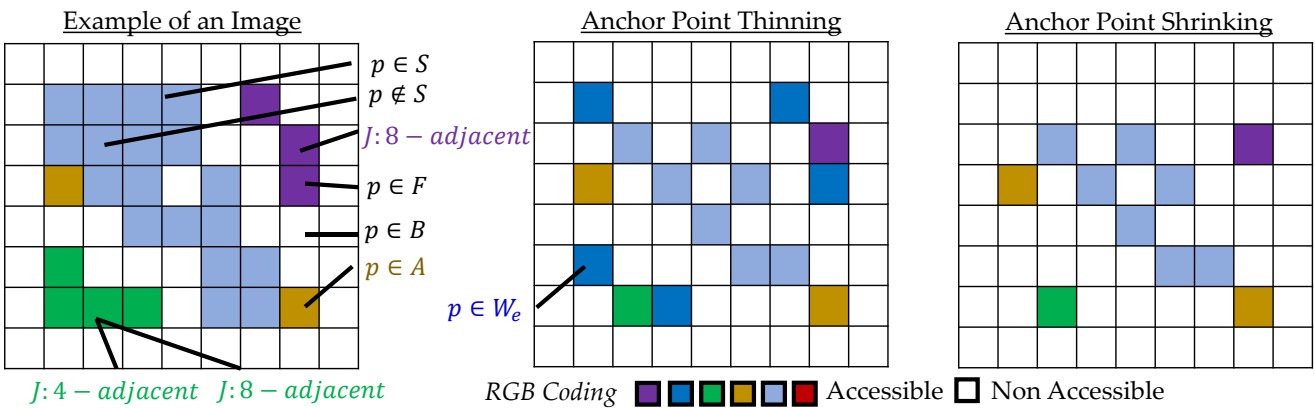

**Figure 2.** Path generation with shrinking and thinning: image with different regions, erosion due to anchor point thinning, erosion due to anchor point shrinking.

To obtain a skeleton, the boundary of $F$ is eroded step by step until only a one-pixel thin line is left. During erosion, the state of a so-called simple point must be checked for each pixel, which can be eroded in terms of topology preservation [45]. A pixel at the boundary of the foreground $p \in F$ is simple and thus erodible if the topological properties do not change after the erosion of this point. The topological properties of the image are preserved if the number of objects, cavities, and holes remain the same after erosion [13,45]. All points that do not meet this condition must not be eroded. This erosion can now be applied iteratively until there are no more simple points. If only the condition of the simple point is checked during the erosion, the erosion method is a so-called homotopy-preserving shrinking [45]. If now additional so-called endpoints $W_e$ are declared as also not erodible, then it is a homotopy-preserving thinning method.

In this work, a so-called homotopy-preserving anchor point shrinkage is chosen for path-finding derived from thinning method of [13], due to its desirable properties of the centered path and topology preservation. Therefore, a simple point condition is checked by considering the preservation of Euler characteristics and the connectivity via an adjacency tree. While the thinning method of [13] uses other shape preservation criteria such as endpoints for the curve skeletons or surface skeletons, this work is restricted exclusively to homotopic shrinkage with so-called anchor points. Anchor points are points $a \in A$ with $A \subset F$ that must not be removed during erosion. Once the process has converged, the skeleton is obtained from the remaining pixels with $S \subset F$. Figure 2 shows a comparison between anchor point thinning and anchor point shrinking of a simple example image. While thinning almost preserves the green and purple pixel chains, shrinking reduces them to a single pixel. If anchor points (dark yellow) are considered, the linkage of the skeleton is guaranteed during thinning and shrinking. The main advantage of anchor point shrinking is the extraction of minimalistic skeletons, that only consider the respective anchor points and the topological paths (see also Section 3.1).

It can be seen that the three objects and the hole in the original image are also present in the two skeletons created with thinning and shrinking. In the following, these anchor points (yellow) are attached to the individual tasks during pathfinding. Furthermore, it is necessary to simplify this skeleton into meaningful path segments, which is discussed in the following section.

### 2.2. Graph Construction with Topological Segmentation

Once suitable paths have been found in the 2D image, a graph can be derived directly from them. For this purpose, a so-called topological segmentation is first performed. Based on the neighborhood condition of the individual points $p_i \in S$, the respective disjoint set of endpoints $W_e$, branch points $W_j$ and skeleton points $W_s$ can be assigned [34]. If a point is 8-adjacent with at least 3 other points, it is classified as an intersection point. In summary, all points $p_i$ from the skeleton $S$ can be classified as

$$p_i \in S \backslash A \begin{cases} |N_n(p_i) \geq 3| \ p_i \in W_j \\ |N_n(p_i) = 2| \ p_i \in W_s \\ |N_n(p_i) = 1| \ p_i \in W_e \end{cases} \tag{2}$$

with $S = W_j \cup W_s \cup W_e$ (see Figure 3) [34]. These different types of points can then be included in $S \backslash A$ to find suitable paths as pixel chains $J_k$. Here, both the first and the last point of the chain are either an end point or a branch point. The points between the start and end points are all skeletal points. Thus, the path can be defined as the set of 8-adjacent points $p_i$ that are ordered with respect to each other as follows

$$J_k = \{p_i, p_{i+1}, \ldots, p_n\} \tag{3}$$

with $\{p_i, p_n\} \subset W_j \cup W_e$ and $p_{i+j} \in W_s$ for $j = \{1, \ldots, n-1\}$ (see Figure 3). Based on that segmented skeleton aa digital twin including the associated information such as the tasks can be derived. For this purpose, the pixel chains are first simplified using a so-called collapse metric. It is ensured that by simplifying the path $J_k$, the distance $d_i$ of a pixel to the simplified polyline $P_k = \{p_{i+1}, \ldots, p_{n-1}\} \subset J_k$ may differ by at most $\epsilon$-pixels (see Figure 3). In this simplification, each step checks that the topological properties are preserved. The following figure shows the segmentation of each point based on neighborhood connectivity, the segmentation of each path based on branch and endpoints, the collapse metric, and the simplification to a graph. Branch points, which are connected to each other (green), are reduced to one point.

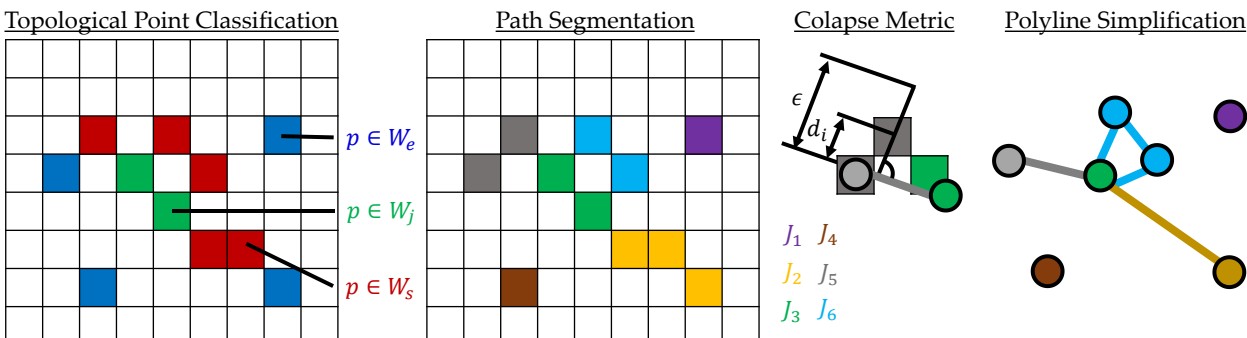

**Figure 3.** Polyline path estimation: topological segmentation of the skeleton, path segmentation, collapse metric for path simplification, simplified path as a polyline.

Subsequently, a graph $G = (V, E)$ consisting of nodes $V$ and edges $E$ can be generated. For the edges of the graph, all information along the paths are used, while the tasks as well as the robots themselves are mapped via the nodes. This graph is now supplemented in the following by the passage width at the respective edges.

### 2.3. Enriching Paths with Local Clearance Width

The passage width can be estimated from the map geometry in combination with the corresponding paths. The skeleton itself has the advantage that it is located in the center of the geometry (see Figure 10). Thus, only the shortest distance from the skeleton to the respective boundary has to be calculated for the approximation of the passage width. This can be achieved with the so-called Euclidean distance transformation. For each pixel, the distance to the boundary is calculated. Then the distance values of the respective path $p_i \in J_k$ can be taken based on the pixels, where the smallest value is the passage width $d_k$ with

$$d_k = \min(\{EDT(p_i), EDT(p_{i+1}), \ldots, EDT(p_n)\}). \tag{4}$$

Figure 4 shows the result of the distance transformation as numerical values, a colored representation of the distance and the intersection of the distance transformation with the individual paths. In the simple example from Figure 4 a minimum width of 1 pixel applies to each path, while in the example from Section 3.2 a variety of passage widths result.

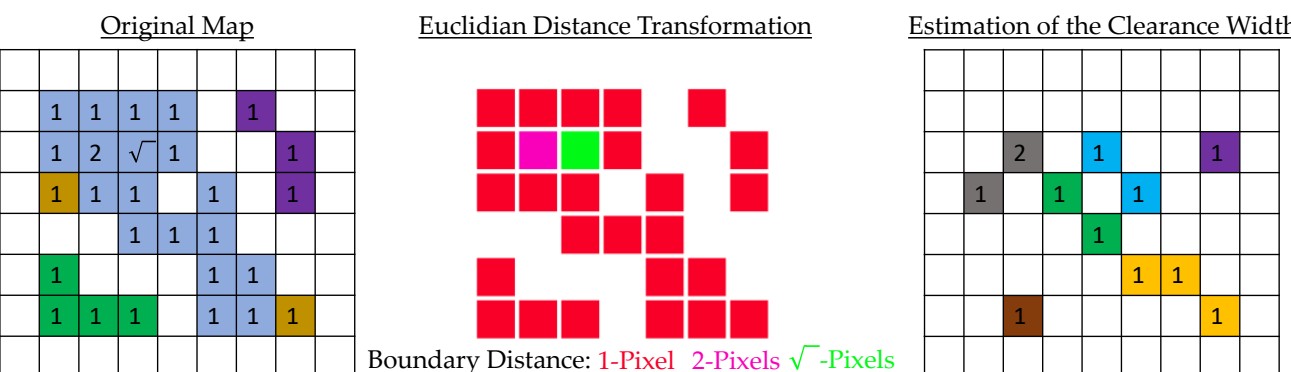

**Figure 4.** Clearance width estimation: map with Euclidian distance values, colored representation, and estimated clearance width of each path.

An edge of the graph can be stored as a tuple with the respective polyline as well as the passage width as $(P_k, d_k)$. If a robot or its load is too large for passing through, an alternative route can be planned directly. Similarly, a path can be traversed by multiple robots, on which the number can be limited by $d_k$. Based on this graph from 2.3 in combination with the clearance width, a digital twin can now be derived, which is covered in the following section.

### 2.4. Deriving of the Digital Twin Using the Graph

Initial conditions are required for the digital twin. These can either be specified by the user or recorded during operation. For example, the duration of the respective tasks or of the traversal of a path can initially be specified via prior information, which is then updated during operation. The speed of the robot can, for example, be embedded in the twin, as rule, depending on the curve radius and the current load, or alternatively determined during operation. Meaningful prior information is optional, but can initially provide helpful path determination.

In our case, the processing time, the possibility to create a certain path between two tasks, as well as the passability of a path, have to be considered in the digital twin. The passability of a path is stored as a Boolean value with (False, True). The time duration in the graph is stored as a numeric value referred to the path and the respective tasks. This travel time of a path can be used to estimate the initial position of the robot along the path, which can also be updated locally during operation. Figure 5 shows the digital twin and the additional information stored. This twin is continuously adapted to the respective robot positions.

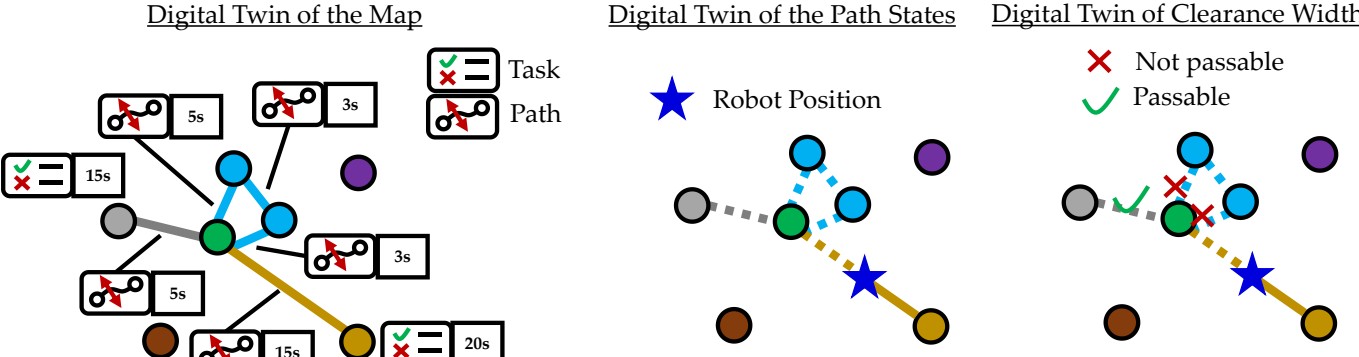

**Figure 5.** Example of generating the digital twin: map with tasks and process time, the position of the robots in the graph, the passable graph for the robot.

It should be noted that the digital twin can be set differently for each robot. If two robots have different maximum speeds limits, the assignment of travel time between the two robots may be different. Such behavior can be added to the system on which the digital twin serves as a reasonable basis. Thus, the digital twin has comprehensive information about the state of drivability, but also local information based on the characteristics of the robot and its load.

With these graphs, an optimization procedure can now be started that proposes optimal trajectories during operation. For example, the Boolean values of the occupied trajectories are considered with the combination of the numerical temporal aspects of the processing time. Likewise, the path generation should react to changing results. In addition to the path recognition in the environment, however, the position of the robot must also be constantly detected and corrected if necessary. For this purpose, it is necessary to track the robots, via, for example a REST-API [46]. In the following, a concept for repositioning the robots based on the presented anchor point shrinkage is presented.

### 2.5. Back to Track: State Update and Position Update of the Robots

If a robot is no longer driving along the path, it can be reordered via an anchor point shrinkage. For this purpose, the pixel position $r_i$ of the robot is included in the set of anchor points $A_r$ for re-tracking. Since the original path should only react to changes in the environment, but not to the local position of the robot, the obtained skeleton of the map without the robot is also chosen as anchor points for re-tracking. Overall, this leads to the anchor points

$$A_r = \{r_1, \ldots, r_n\} \cup A \cup S. \tag{5}$$

If a new skeleton $S \subset S_r$ is found, the paths for repositioning $J_r$ can be found with $J_r \subset S_r \backslash S$. These paths can be simplified to polylines analogous to Section 2.2 and given clearance widths analogous to Section 2.3. Figure 6 as well as figures in Section 3.5, show several examples of a repositioning considering the anchor points. In particular, in Figure 6 a robot deviation is marked as a gray pixel, which leads to an additional segment and a corresponding calibration.

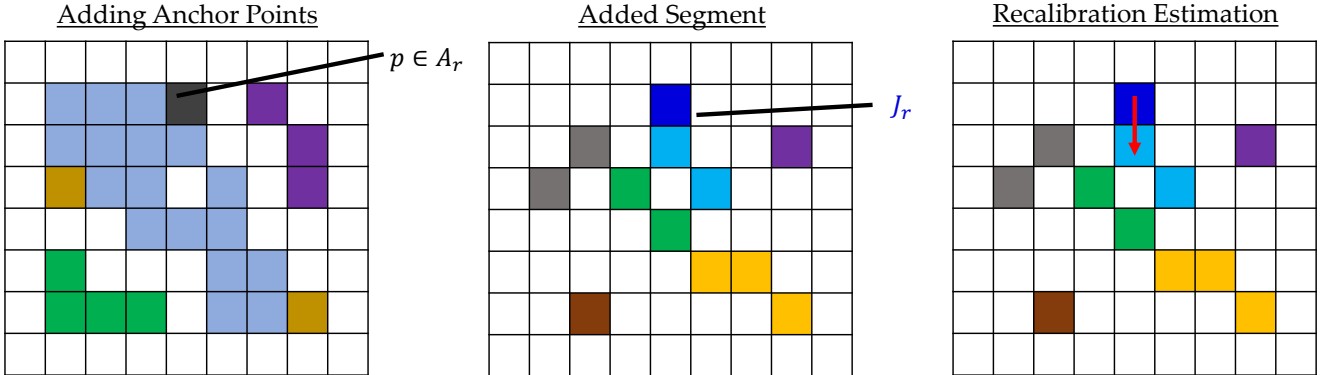

**Figure 6.** Anchor points for recalibration: adding robot position as an anchor point, additional segment during constrained shrinking, recalibration to reduce the additional segment.

In this work, however, repositioning for calibration does not affect the structure of the digital twin. Since the position of the robot changes constantly, but the trajectories usually only adapt when the environment changes, a certain uniformity is maintained when the trajectories and tasks in the digital twin are considered exclusively. The following covers the architecture of controlling iRobot Roomba 981 which can be accessed via representational state transfer application interface [46].

### 2.6. Implementation Details on Updating the Digital Twin

The main architecture chosen is a microservice architecture with a REST interface. The communication between the robot and the server is performed via [46] and the communication between the server and the digital twin is performed via a Python interface. This interface converts the map into an RGB map and identifies the current position of the robot. Figure 7 shows the architecture used in the example of a vacuum cleaner robot and the operation of the generated digital twin (no generation, only updating).

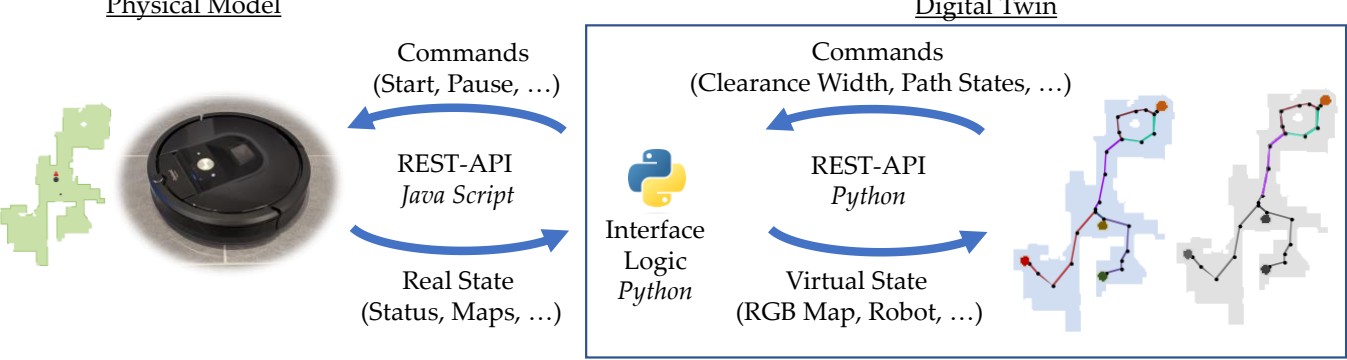

**Figure 7.** Use of the digital twin: interface to the physical robot, interface to the digital twin.

The robot position can now be updated directly via the above interface. Likewise, new obstacles or new paths can be updated directly as a new RGB map. In the above implementation, the individual tasks are fixed with an X-Y coordinate so that they can be transferred to a new RGB map. The Python module can now be used to decide when to update the entire digital twin (regenerate paths) or only adjust the position of the robot

along the path. This decision finding will be further elaborated as part of future work. Therefore the tasks will serve as an indicator for image registration (assembly of several maps to one).

In the following, the described procedures from 2.1 to 2.5 will first be applied to a single, but a much larger map. Furthermore, the described procedure is also applied to a large number of map variants as well as changes in the map or the off-track position of the robot.

## 3. Results

The respective approaches from the previous sections are now tested step by step on an example with several tasks as well as robots and possible paths in the corresponding chapters 3.1 to 3.5. Then, the whole process is evaluated again on a variety of different map layouts for virtually generated complex 2D maps in 3.6, for virtually generated 3D maps in 3.7 and for 2D real world map provided by iRobot Roomba 981 in 3.8. Figure 8 shows a map with five tasks, a gray drivable area, several different colored obstacles and 3 different robots (agents). The size of the robot is represented by the size of the circle (purple). The largest robot cannot drive on the small path in the upper left corner of the image.

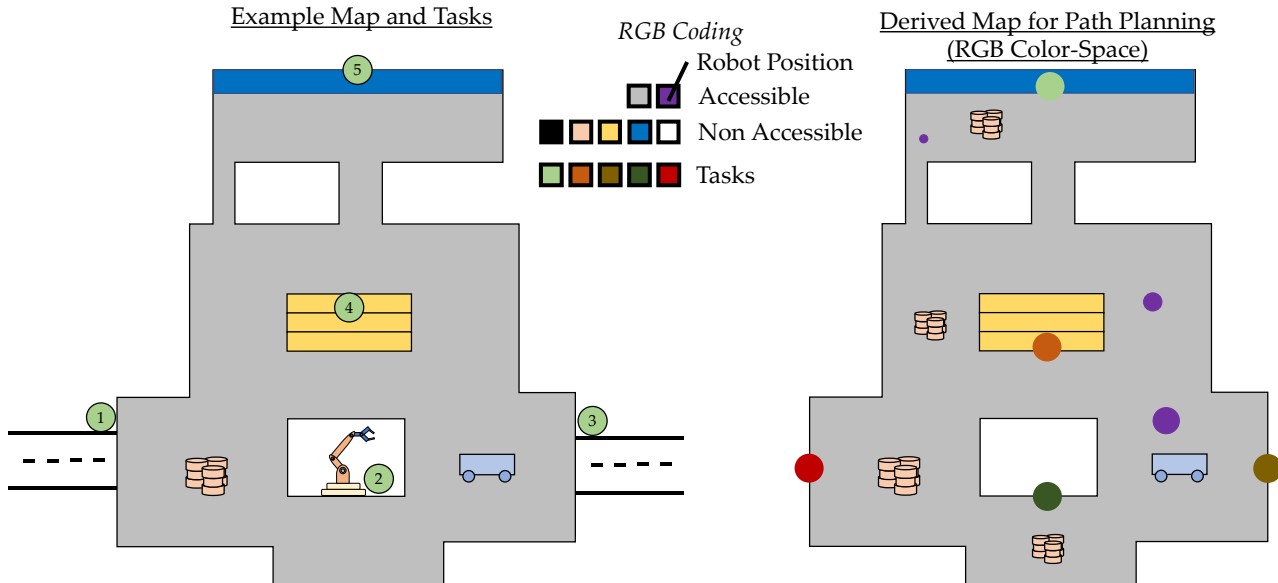

**Figure 8.** Task 1-5, robots, and paths based on RGB map: original map, derived RGB-coded map.

The map is coded as an RGB image. Here, the different tasks are each marked with their own color. In the following, the skeletonization described above is applied to this map.

### 3.1. Constrained Path Estimation with Homotopic Shrinking

Figure 9 shows the transformation of the map into a traversable area (yellow) and a non-traversable area (black-blue) as well as the individual tasks in green. Since the resolution of the map is very high, the map is topology-preserving reduced by a factor $1/4$ to a more suitable resolution. Therefore, it is ensured, that the rescaled image preserves the same number of holes, objects, and 3D cavities in comparison to the original image.

Based on this coded map, the method of skeletonization is now applied. Figure 10 shows the result of a classical shrinking, a thinning with anchor points and the proposed method of shrinking with anchor points.

While shrinking the maps leads to topology-preserving paths, it lacks in linking to the individual tasks. The path suggestion with anchor point thinning can link the individual tasks, but also generates paths (cyan circle) that are not useful (low robustness). Anchor point shrinking shows appropriate minimalistic paths for the robots to access each task. This method leads to a high robustness and ensures that every possible path is captured.

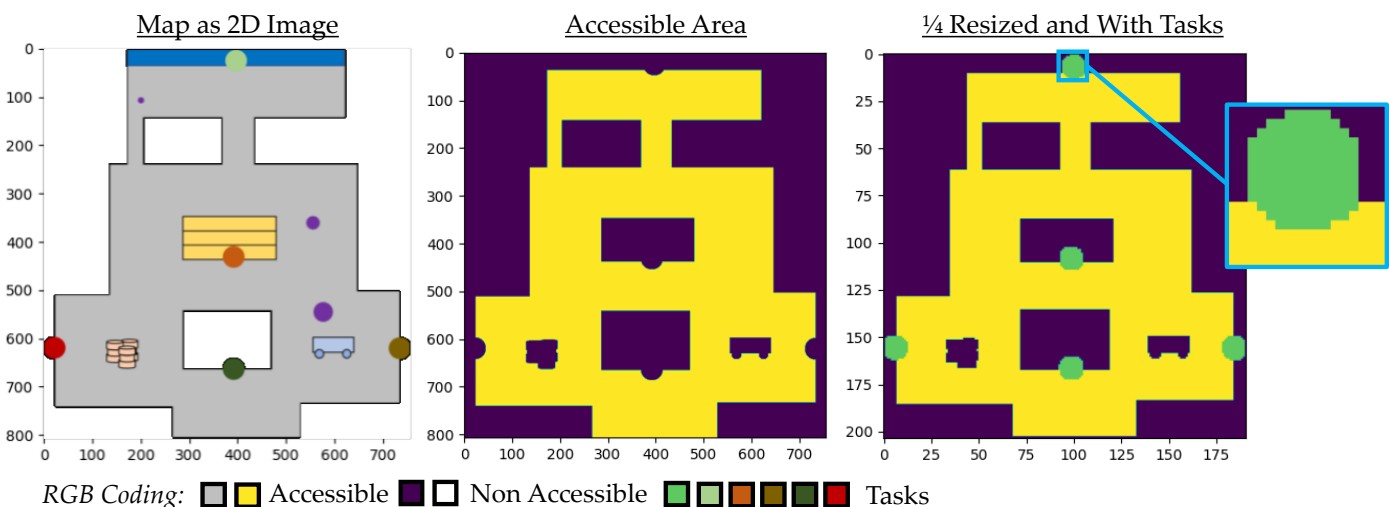

**Figure 9.** Segmentation in the foreground (yellow) and background (dark purple): RGB map, map with foreground and background, map with additional tasks (green).

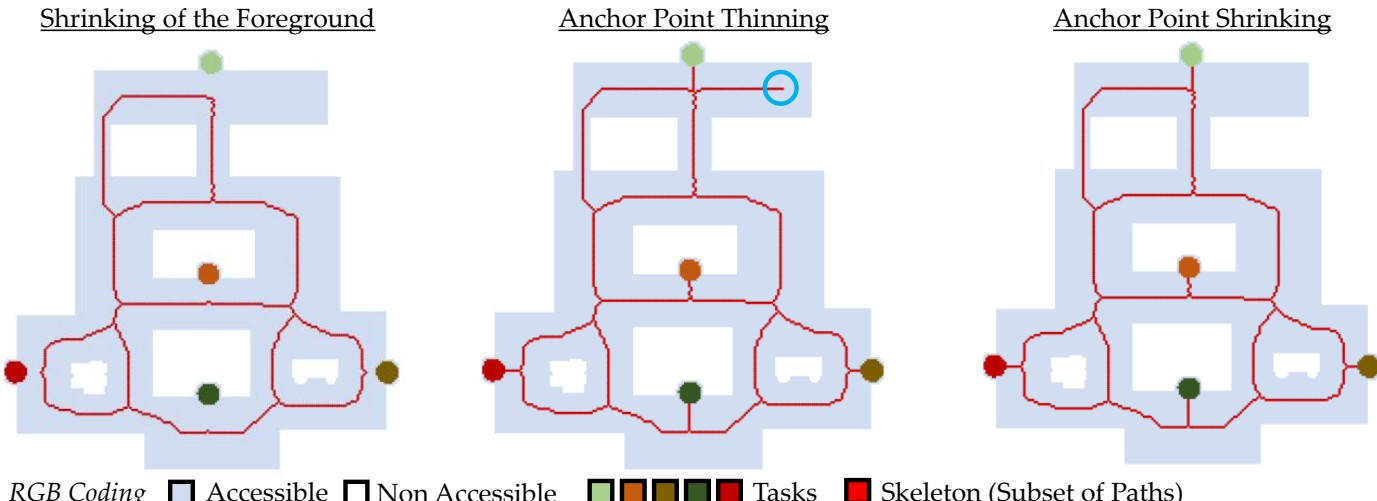

**Figure 10.** Path generation by homotopic erosion: use of shrinking, anchor point thinning, anchor point shrinking.

The path generation can now be compared with directly applying path algorithms such as A* without environment simplification. Since the classical A* only determine a low-cost path from a source to a destination, a simpler map layout with only two different tasks is chosen for comparison. Figure 11 shows the solution of the path algorithm for two different combinations and the reduced path suggestion using anchor point shrinking.

The path of the A*-algorithm is close to the boundary and directly touches the corner. The decision of which paths are more favorable must be held based on the use case. For autonomous systems, a robust and predictable path sequence is more important than choosing the shortest path, especially in an autonomous factory or healthcare environment. Humans will know as prior that robots will always head for the center of the room and stay as far away as possible from non-accessible areas. Based on these paths, the digital twin can now be derived directly. In the following, the graph is automatically generated from this map.

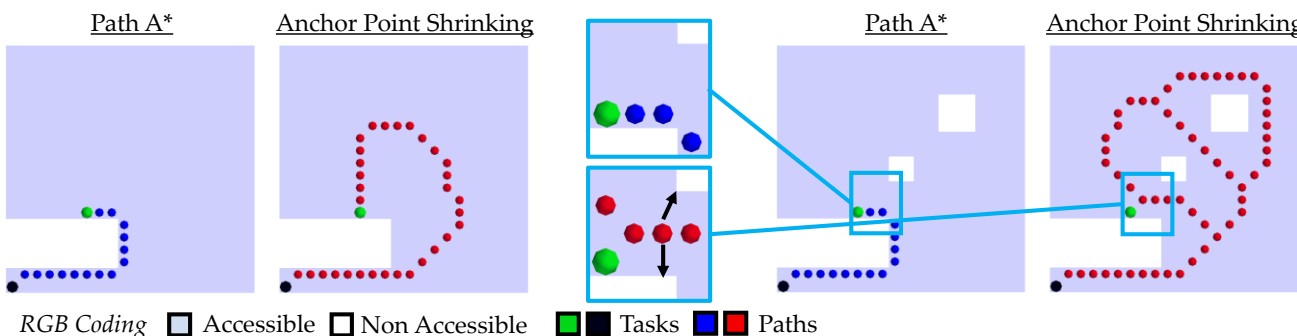

**Figure 11.** Comparison of applying A* on the image environment and a reduced subset of paths considering anchor point shrinking: first example, second example.

### 3.2. Graph Construction with Topological Segmentation

The skeleton can now be segmented based on the branch points as well as the respective connection to the anchor points and then reduced to polylines. In Figure 12, both the segmentation of the paths and the simplification as a graph structure are shown.

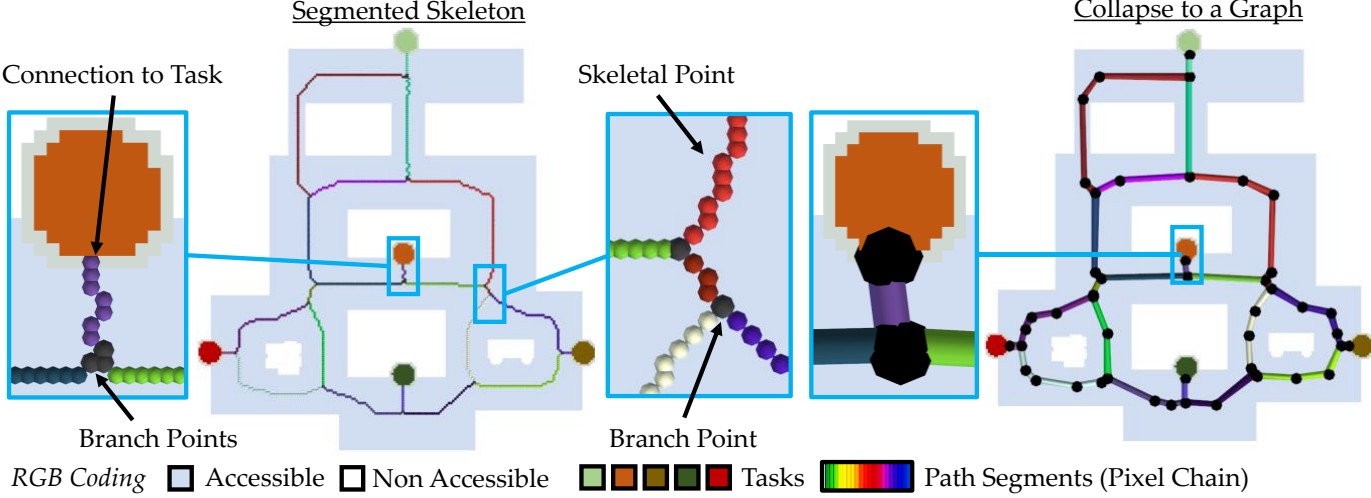

**Figure 12.** Simplification of paths in a graph: topological segmentation, simplification to polylines.

This graph can now be enriched with additional information. Since the individual tasks are coded with different colors, the duration of the respective task can be stored directly. Another important piece of information is the local passage or clearance width. This makes it possible to determine, which of the differently sized robots can pass the respective path at all. In the following, this information is automatically obtained from the map layout.

### 3.3. Enriching Paths with Local Clearance Width

Figure 13 shows the calculation of the passage width by means of the so-called distance transformation, the estimated paths, and the subsequently determined passage width. This passage width is visually represented as circles around the respective path. The brown path shows a very small passage width whose path cannot be traversed by the largest robot.

The passage width can be estimated using the distance transformation. Since the minimum distance value is selected as a criterion for each pixel of the respective path, the most conservative solution can always be determined automatically. Based on this passage width, the robots can now be controlled in a way that, for example, several robots are allowed to occupy a path, or robots with a large load are not allowed to approach certain paths. In addition, information can be obtained directly as to whether a certain load can be moved from task 1 to task 5. In the following, the behavior of the moving robots in the

selected example map is shown in addition to the travel width. This information can now be collected in a digital twin.

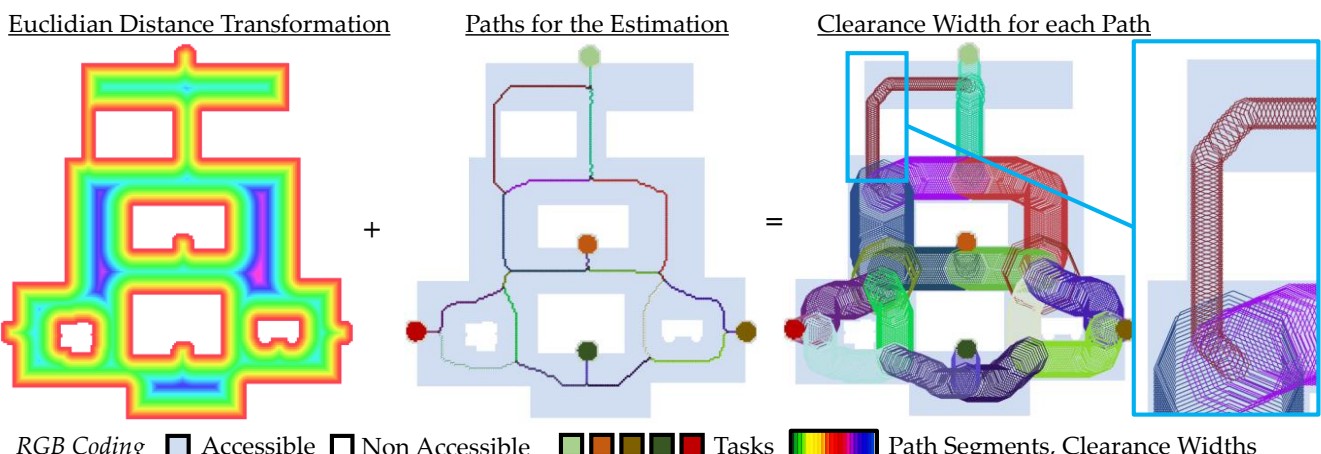

**Figure 13.** Minimum clearance width estimation: Euclidean distance transformation, segmented paths, passage width of each path.

### 3.4. Deriving of the Digital Twin Using the Graph

In this case, the initial travel time as well as the duration for processing a task is integrated as user input. The path width is assigned to the respective edges. Figure 14 shows schematically the assignment of the properties of the respective paths and tasks. Likewise, the application of such a digital twin is illustrated, in which the individual position of the robot on the path as well as the possibility of driving on a route is mapped.

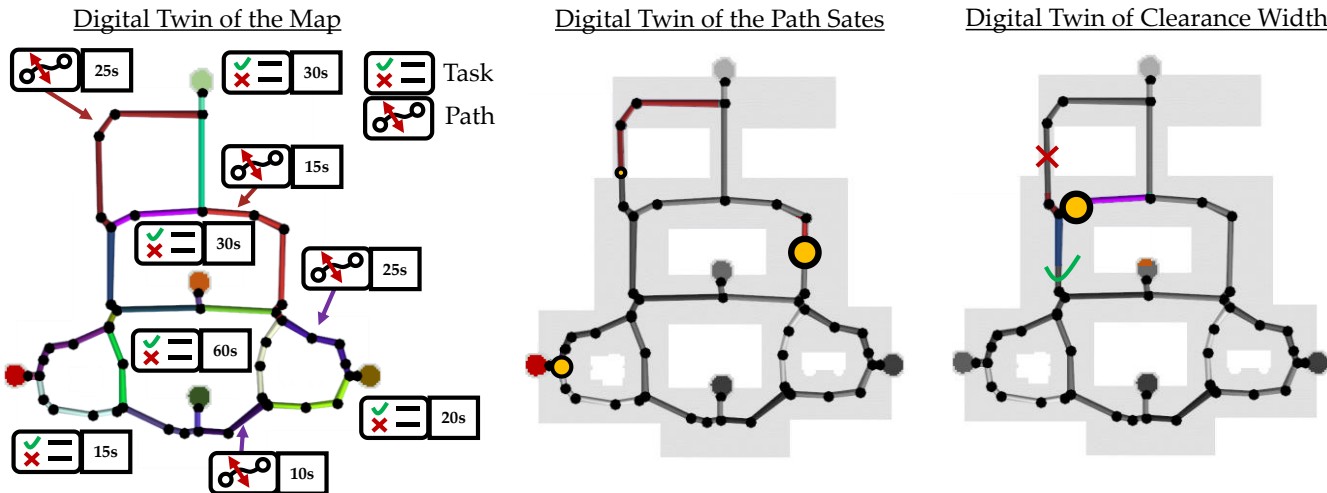

**Figure 14.** Generation of the digital twin: labeling of edges and nodes over task and process time, a model for traversing paths, a model for passable paths.

In addition, on creating the twin, the following section covers the reposition of incorrectly positioned robots. These must be returned to the existing paths.

### 3.5. Back to Track Update

On the selected map, the start positions of the robots are located next to the paths. These robots must now be automatically steered back to the original path. Figure 15 shows the repositioning from 2.5 on the map. A suggested path (blue dots) can be determined automatically on which the robot can be moved back onto the path.

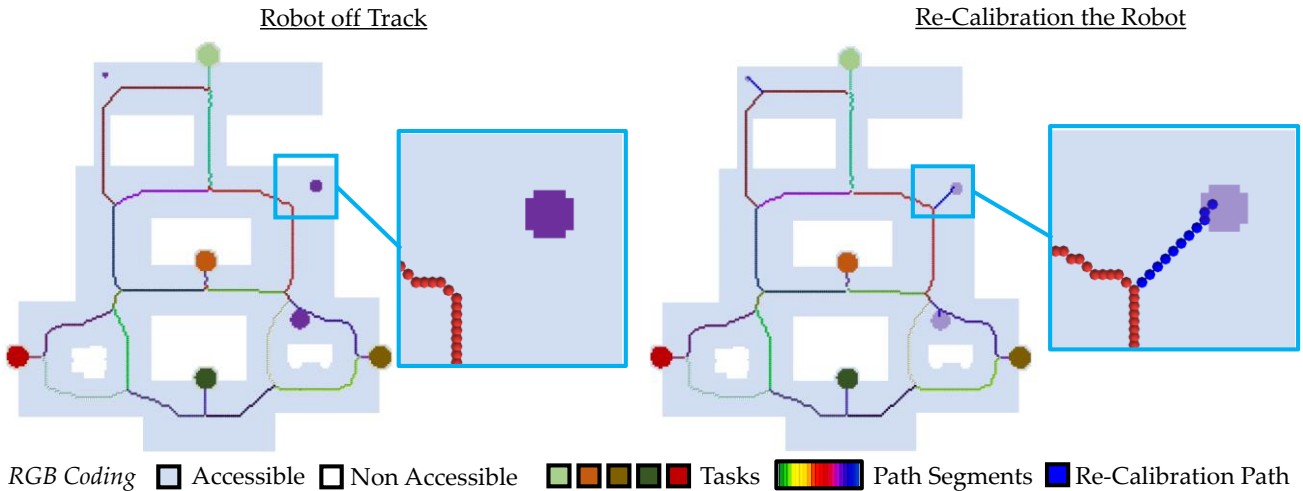

**Figure 15.** Repositioning the robot: map with the robot not on track, repositioning path for the robot.

This concept can also be used during an operation to constantly update the respective robot positions. It also shows that all robots can be handled simultaneously with one repositioning step. This automated generation of the twin as well as the repositioning of the robot position can now be used in a changing environment. In the following, the chosen approach is tested with different off-track positions, the sudden appearance of obstacles, and alternative map layouts of generated virtual 2D maps.

*3.6. Changing the Map and Its Obstacles*

In the following example from Figure 16 new obstacles (cyan circles) are integrated into the map. An update of the paths shows an automatic change of the paths including the detection of the passage width.

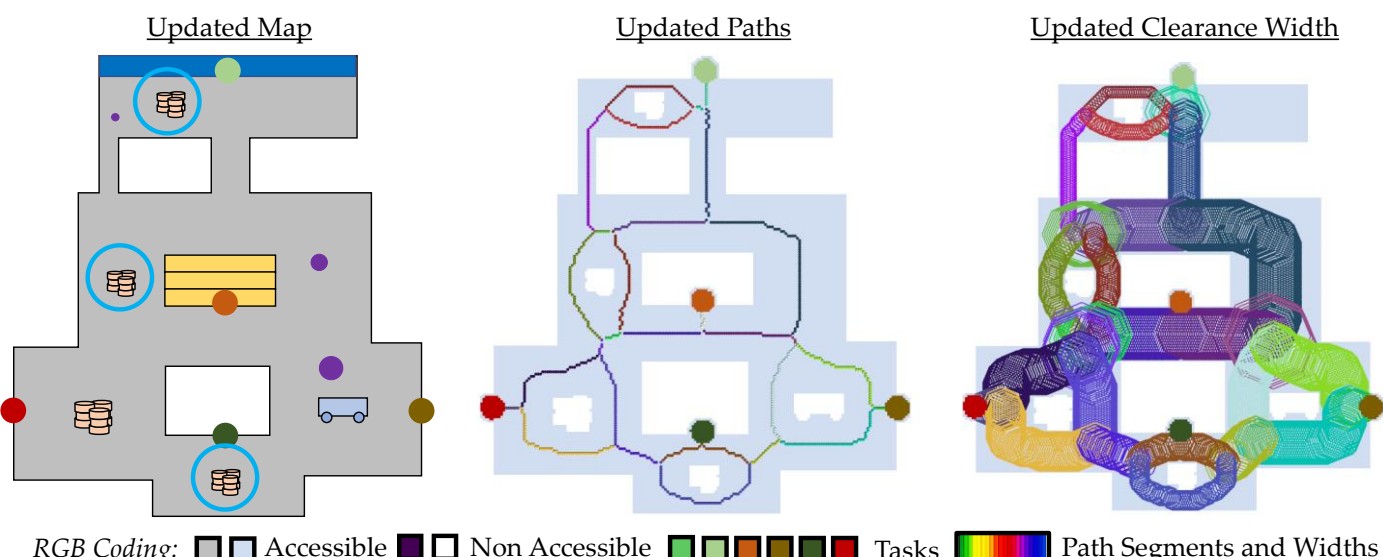

**Figure 16.** Adding new obstacles on the map: RGB map, newly generated paths, newly estimated clearance width.

The new paths are located further in the center of the geometry and can be built as a digital twin as explained in Section 3.5. All newly created paths are also represented topologically so that in addition to the suitable short paths, the alternative possibilities are calculated. In addition to the appearance of new obstacles, the map itself can also be extended. Figure 17 shows the path generation of an extended map with new tasks, obstacles, and robots.

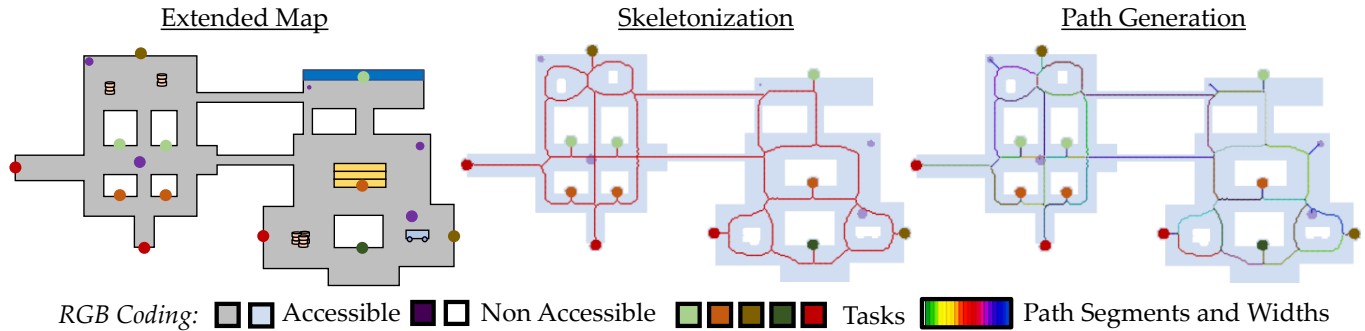

**Figure 17.** Map extension: RGB map, skeletonization of the extended map, path generation of the extended map.

Likewise, the digital twin, including the clearance widths, for the new area is automatically acquired. Figure 18 shows the digital twin, the Euclidean distance transformation, and the additional clearance width of the new path.

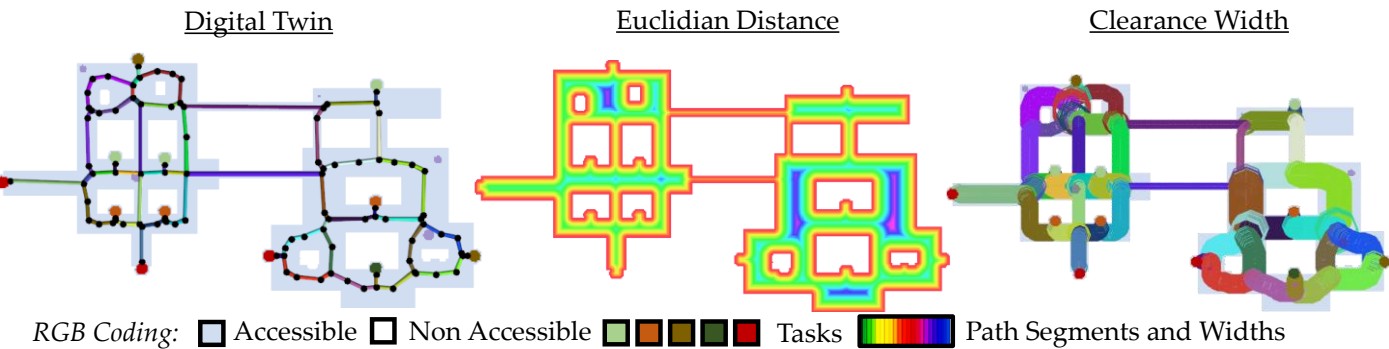

**Figure 18.** Digital twin for the extended map: graph generation, Euclidean distance, clearance width.

In addition to expanding the map and placing obstacles, path generation can also be analyzed for noisy input images. In the map below, there are 1-pixel holes in the traversable area. These holes can be removed in a preprocessing step, but for robustness analysis, they should be left unchanged. The generated paths in Figure 19 avoid the holes as they topologically contribute to the initial image.

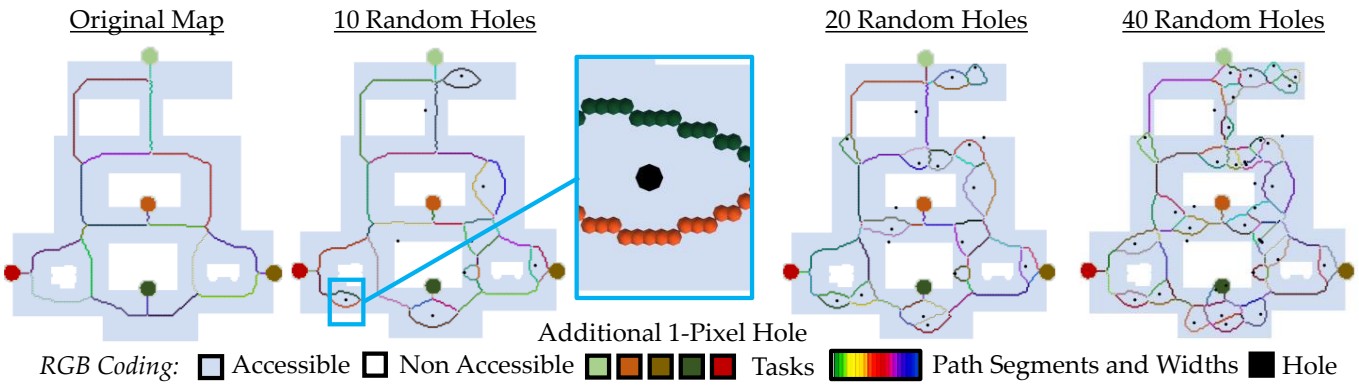

**Figure 19.** Noisy map: path generation, Euclidean distance transformation, clearance width.

The results show that noise-free images should be applied for path generation. Otherwise, as shown in Figure 18, numerous branches are created, complicating the resulting digital twin. However, due to the robustness of skeletonization, paths can be found that accomplish the task while avoiding individual holes. Therefore, this method is very well suited to respond to changing inputs. While the map layouts in Figures 17 and 18 are very

simple, Figure 20 shows the application of path generation using a maze as an example. There are several tasks, a robot, and a blocked path.

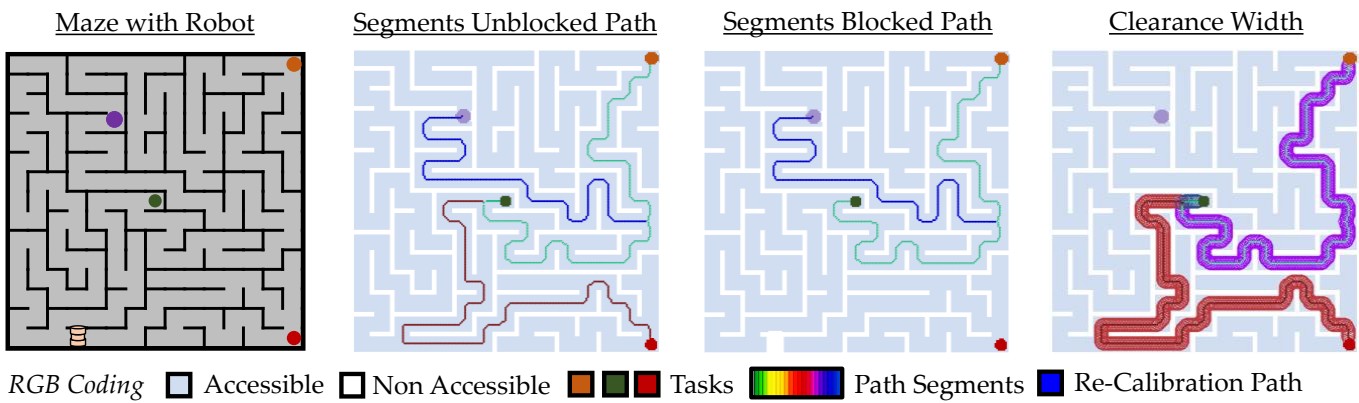

**Figure 20.** Path generation of a labyrinth: RGB map, path generation without blockade, path generation with blockade, clearance width.

The paths including the path widths can be determined automatically. If a path is completely blocked, no path is created. It also can be determined automatically whether a task can be reached at all. The same procedure can be used for a more complicated maze. Figure 21 shows the path definition, the graph for the digital twin and the Euclidean distance transformation of the image for a more complicated maze.

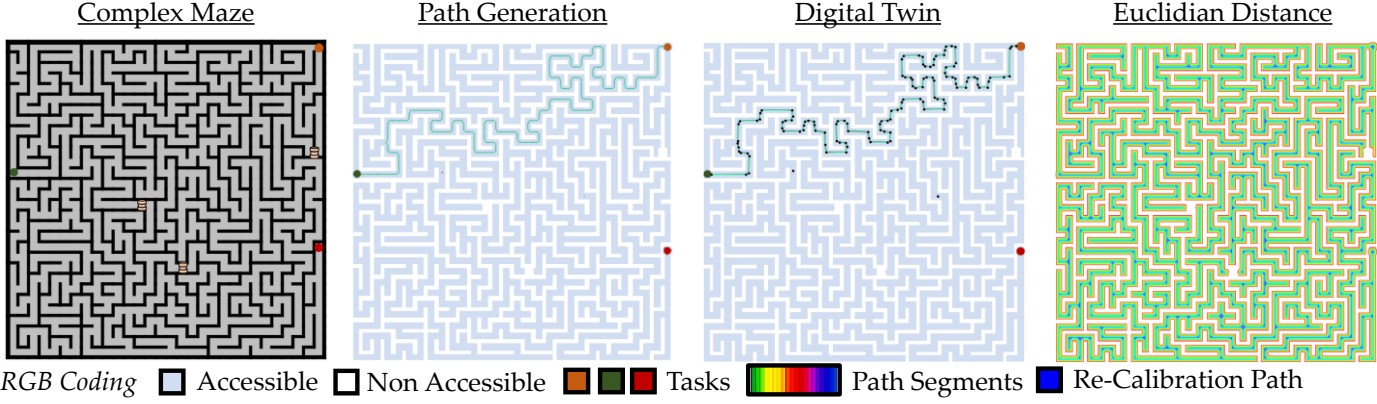

**Figure 21.** Path generation in a complex maze: RGB map, path generation, the graph for digital twin, Euclidean distance transformation.

The path to the red marked task has been blocked so that the is completely unreachable for all other tasks.

### 3.7. Digital Twin Generation in 3D Virtual Environments

In addition to the use of 2D maps, 3D problems can also occur in various domains. For example, in a factory, there are ramps or bridges including underpasses whose topology cannot be captured in 2D. Figure 22 shows the generation of paths for the 3D problem.

It has been shown that the generation of a digital twin by skeletonization can react directly to changing circumstances. Thus, new obstacles and tasks can be recognized implicitly. Likewise, alternative routes can be found, and 3D maps can be generated. While in Figure 22 the drivable area is modeled by flat 3D surfaces, so-called flight zones and drivable zones can also be considered as3D images. Figure 23 shows a simple 3D model with a robot and a helicopter. The drivable region is captured as a polygon mesh that can be rasterized into a 3D image. The individual volumetric pixels (voxels) are then labeled with their respective tasks and positions of the robot. The distance transformation must

now be generated based on the 3D object, so that the height can be considered in addition to the length and width.

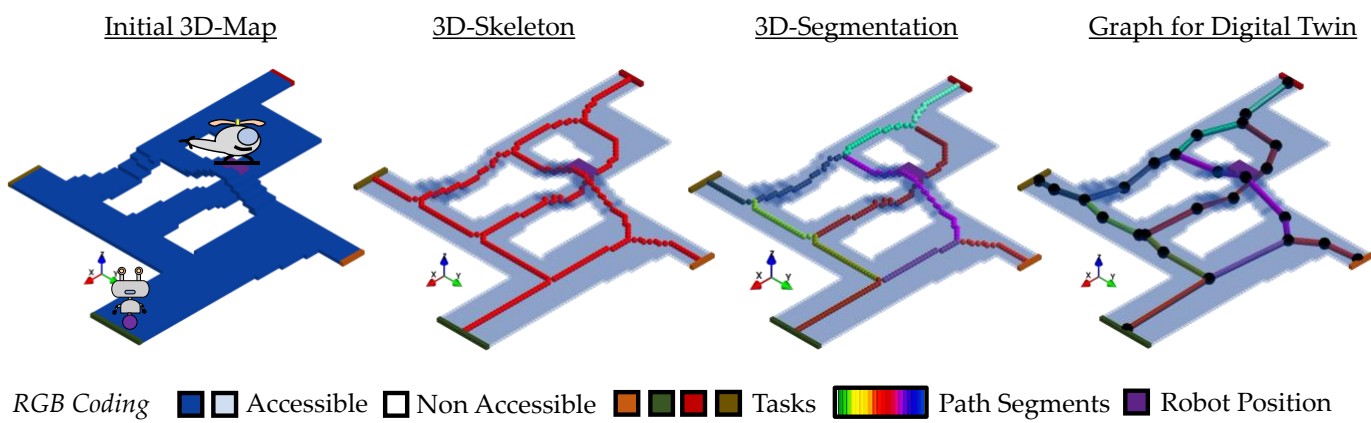

**Figure 22.** Generation of graphs for 3D maps: 3D RGB map, the skeleton of the 3D map, the segmentation of the 3D skeleton, the derived graph for the digital twin.

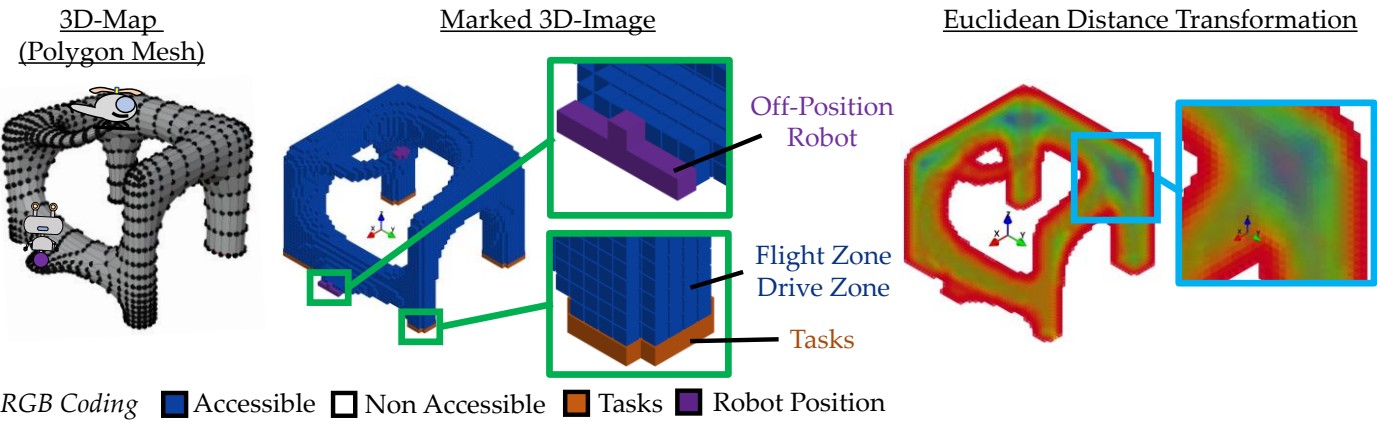

**Figure 23.** Drone and robot example with 3D flight and 3D drive zones: polygon mesh as flight zone and drivable zone, marked 3D map, Euclidean distance transformation of the flight and drive zone.

Based on the 3D image, a skeleton can be created and further segmented (see Figure 24). In addition, individual robots or drones can be re-positioned to the paths. Finally, the digital twin can be obtained in the form of a graph by using the distance transformation and the generated paths.

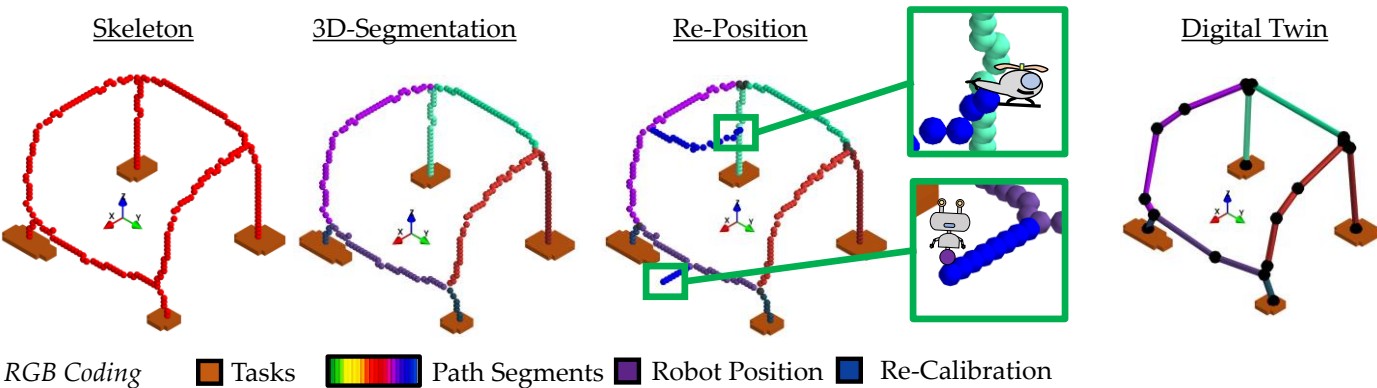

**Figure 24.** Generation of graphs for the 3D drone/robot sample: the skeleton of the 3D map, segmentation of the 3D skeleton, re-position of the off-track drone and robot, derived graph for digital twin.

This digital twin can also be used to consider the paths based on the individual robot properties. For example, the mobile robot on the ground can only traverse one path and thus only one pair of tasks, while the drone can traverse all tasks and paths. This additional information can be processed after path creation or already in the 3D model.

### 3.8. Comparison of Different Path-Algoirthms

To compare the proposed path finding with alternative methods, the 2D map from [24] is redesigned and added with the path provided by the skeletonization method in combination with A*. Figure 25 shows the original map, the simplified environment, the clearance width of each path, and the comparison of the paths with alternative path finding methods.

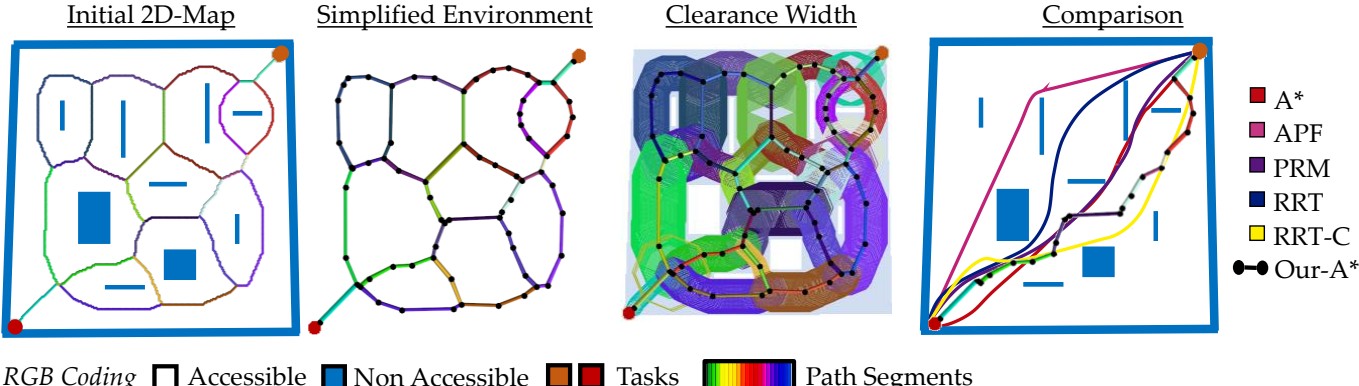

**Figure 25.** Comparison of the simplification and path suggestion considering the use-case of [24] with alternative path finding algorithm.

In contrast to the classical path finding methods, the found path is maximally far away from the obstacles. While A* calculates in the example of [24] the shortest path of the described methods, it touches the boundary next to the obstacles. Such behavior also occurs in APF, PRM, RRT, and RRT-C, which is due to the too free choice of the environment. In the following, the method is applied to maps from real environments.

### 3.9. Digital Twin Generation in Real World Environment Using a Vacuum Cleaner

With robots that have suitable sensors, a map can be created, that can then be evaluated in a virtualization environment such as Gazebo. However, to avoid the additional cost of a dedicated system for care facilities, this work creates a map using a conventional vacuum cleaner robot from iRobot Roomba 981 which can be accessed via REST-API [46]. The map can be directly retrieved from the API of [46] and selected for the construction of the digital twin. Furthermore, such API allows the access to directly control the robot via LAN or Cloud. Figure 26 shows two different maps, the manual definition of the tasks, the automatically determined robot position, the segmented paths and the respective path widths. The white holes in the map itself indicate obstacles on the path that can be detected directly in the digital twin.

The different use cases demonstrated the generation of a digital twin for virtually 2D and 3D environments as well as for 2D maps by scanning a real environment by the REST-API [46] for the vacuum cleaner robot iRobot Roomba 981. Overall suitable digital twins could be obtained for all examples. By using topological thinning, properties such as task connectivity as well as topological alternate path configuration could be ensured. In the following, both the advantages and the disadvantages of the described method are discussed.

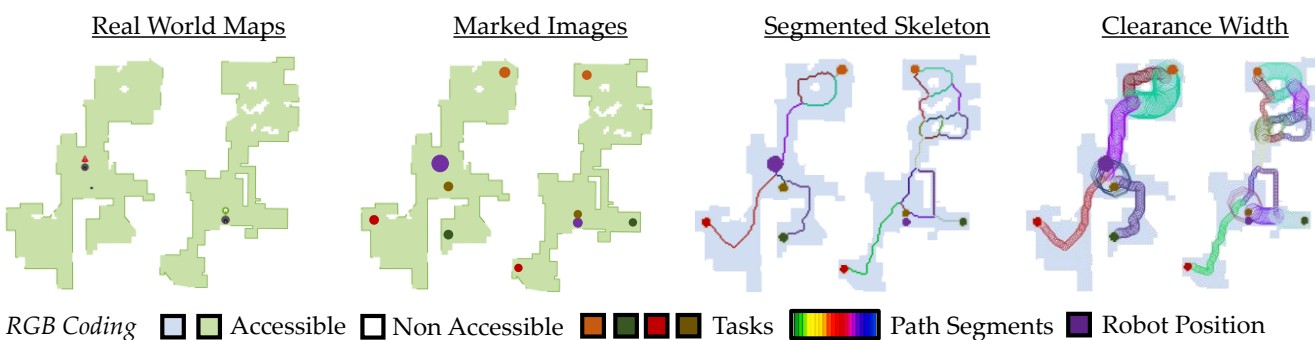

**Figure 26.** Generation of graphs in a real world use case: two maps, two marked images, two segmented skeletons and two path suggestions including clearance width.

## 4. Discussion

Path finding in 2D and 3D maps is one of the most common tasks in robotics, inPC games but also in route planning. In addition, on completing individual tasks, the position of each agent in the environment must also be considered. A so-called shrinking provides a opportunity to capture topological paths that ensure linkage to tasks when possible.

### 4.1. Path Generation

Since the method is based on the acquisition of image data, a digital image is required as the initial environment. The provision of such a map can be generated, e.g., by hand sketches, scans, e.g., by LIDAR but also by manual scanning. Since noise can often occur, especially when scanning the drivable surface (see Figure 25), a robust filter must be available to clean the environment. For example, if there are individual 1-pixel holes in the map, these are assumed to be holes and are considered topologically during path generation. It was shown in Figure 18 that a noisy map with a large number of holes can lead to small individual paths. It is therefore necessary to clean up the map beforehand using suitable filtering methods or to manually specify a clear layout.

If suitable maps are available, the paths can be estimated fully automatically. In contrast to the alternative path algorithms, A* or Dijkstra applied on the whole environment, topological reduced paths with alternative routes can be considered by anchor point shrinking implicitly (see Figure 11). These additional paths can be used to avoid an obstacle or a hole on two sides. Since path generation is mainly determined by the traversable area, only the paths to each task change locally if additional obstacles occur also locally (see Figures 16 and 17). The paths to avoid new obstacles change only locally, resulting in robust routing and human-expectable behavior. For example, in A* on the whole environment, the path layout changes greatly when the position of the target or task also changes (see Figure 11). This can lead to unexpected behavior, causing the agent to fidgety implement the path changes.

In path algorithms such as A* or Dijkstra the shortest paths are determined by cost functions and the paths are often placed near the corners of the traversable area (see Figure 11). However, since the position of the robots is subject to some uncertainty, they may cut off the possible non-traversable area. In a factory or health environment, this leads to visually confusing situations and boundary violations. Similarly, there is often insufficient space to correct inaccurate robot positions. In addition, trajectories from the A*-method can lead to unintuitive trajectories, such as when the robot crosses the lane from the left edge to the right edge, leading to a possible overlap with the trajectories of oncoming traffic. Path reduction and generation based on the skeletons result in centered paths so that the greatest possible distance from the boundary of the drivable area can be realized. This robust and expectable path generation can be used to train humans in factory areas as well as in care stations to behave in such a manner. When a robot needs to be bypassed, the staff or patient can move directly to the boundary of the traversable area. Since such behavior is also found in road traffic in the form of a drivable road in

the middle and pedestrian paths at the boundary, this is a familiar and intuitive behavior. Similarly, the path with the largest distance to the boundary provides the opportunity to reposition deviating robots with sufficient distance. The main drawback of skeleton-based path simplification is that the paths are often not efficient. While A* and Dijkstra without restriction aim for the shortest possible paths, the paths from the anchor point shrinking show suboptimal paths in terms of path length. For example, if process time has to be optimized, path generation via A* on the whole environment would be the better choice.

### 4.2. Generation of the Digital Twin

Once a path is found, it can be used to build a graph that serves as the basis for the digital twin. In the examples from Section 3, robust derivations of the twin were shown. For this purpose, the paths were supplemented with additional information such as the automatic path width or the manual specification of turn times. A change in the environment shows a change in the digital twin. Especially worth mentioning is the locality of the process. For example, if a new obstacle is suddenly found on the map, the paths and the robot commands only have to be adjusted locally. As a result, all unaffected robots can perform their task without modification. In addition, the path width specification offers the possibility to let opposing robots drive (left, right) if they can drive side by side including the size of their payload. Based on this twin, a state diagram could be created to evaluate drivability. Likewise, based on this, the position of the robots along the paths can be directly recorded. If repositioning is necessary, the individual paths for the paths of the robots can be generated locally without affecting the global paths of the digital twin. This combination of robust path generation, but also the possibility of paths for repositioning that do not affect the original paths, results in a robust twin. Since bridges, overpasses, or underpasses may also be present in the application, the procedure in Figures 22–24 were extended to 3D. The results show reasonably automated path generation even for 3D maps. However, the biggest drawback is the need for accurate RGB coding in the map and an accurate map layout. In real-world applications, a variety of artifacts can be caused by different sensors. In addition, these errors are not accounted for in the procedure described above, so further approaches to filtering the errors and capturing multiple maps has to be pursued in the future.

In summary, path generation using anchor point shrinkage in combination with distance width allows the construction of robust digital twins. Local changes such as people on the roadway or repositioning of a robot also affect the map only locally. This results in paths that are centered, which provides maximum space for recalibration, but can also result in predictable driving behavior similar to rules such as on the roadway.

### 4.3. Outlook

Real-world applications can now be investigated in future work. Appropriate pre-processing can be used to remove small holes and irregularities. This may include selecting skeletons from shrinkage to identify, remove, or connect small holes or island-shaped areas.

Furthermore, such digital twins can now be enriched with an uncertainty of the robot's position or even the kinematics of the local robot and drone itself. By constructing digital twins for each robot and the individual tasks themselves, a more detailed environment can be set up. In such an environment, the twins of the task, robot and the path generation communicate and influence each other.

The digital twin itself can now be enriched with further information on the basis of the graph, which is oriented towards the areas of application, e.g., in a factory or a hospital. For example, patients, visitors and staff can also be equipped with sensors that are recorded in the twin. If the staff needs to intervene quickly or a robot needs to clean the floor, the nearest staff or robot can be paged to reach the patient.

## 5. Conclusions

A digital twin describes the virtual representation of a real process. This twin is constantly updated with real data and can thus control and adapt the real model. Designing suitable digital twins for path planning of autonomous robots or drones is often challenging due to the large number of different dynamic environments and multi-task and agent systems. Generating robust paths specifically on controlling a variety of robots and tasks is one of the most common problems in controlling autonomous systems. Although there are many different route planners, often only criteria such as the shortest distance are considered.

In this article, a shrinking-based method is presented to reduce the environment to a reasonable subset of paths. In particular, the robustness and implicit advantages of shrinkage for path generation are presented. The main advantage in contrast to the shortest path is the topological ensurance of all alternative paths that are maximally far away from the boundary (centered). This centrality provides the advantage that the paths are predictable and sufficient for recalibration. The alternate paths also provide the ability to avoid a route if it is already occupied.

In addition, digital twins can be derived directly from these paths that can be used to reference both the individual robot states and the paths. To this end, a path simplification was used to generate a graph from the digital image in which the individual task time is referred to the nod of the graph and the individual path time and path width is referred to the edge of the graph. This twin can now be used to control multiple autonomous robots in future works.

The variation of map layouts as well as the placement of individual obstacles and the use of a 3D map demonstrate robust pathfinding that can handle multiple tasks simultaneously. Changes due to obstacles only affect the associated paths locally, leaving the other paths in the twin unaffected. Therefore, the control commands for the robots only need to be updated when new obstacles appear locally. If a robot is incorrectly positioned, it can be steered back to the nearest path via an anchor point shrinkage that is decoupled from the digital twin.

In summary, this article shows the generation of a digital twin, enriched with process times as well as path widths, whose paths are constantly centered and which can react to a changing environment only with a local change in the path layout.

**Author Contributions:** Software, conceptualization, methodology, visualization, validation, investigation, resources, data curation writing—original draft preparation: M.D., writing—review, conceptualization and editing: H.V., P.S., S.G. and S.B.; project administration, funding question, supervision: S.W. All authors have read and agreed to the published version of the manuscript.

**Funding:** This research received no external funding.

**Institutional Review Board Statement:** Not applicable.

**Informed Consent Statement:** Not applicable.

**Data Availability Statement:** Not applicable.

**Conflicts of Interest:** The authors declare no conflict of interest.

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
