# Peer review of "Generating Digital Twins for Path-Planning of Autonomous Robots and Drones Using Constrained Homotopic Shrinking for 2D and 3D Environment Modeling"

_applsci, doi:10.3390/app13010105_

Round 1

Reviewer 1 Report

the paper seems interesting and the topic is updated in the field of path planning. but, in my opinion, it is necessary to check the following remarks:

1- the decision-making process should be considered and defined by comparing the results.

2- the analytical analysis should be added to the abstract and conclusion sections

3- the case study with precise results for a real-world system should be selected.

4- the refs should be covered the all planning process in the introduction sections and covered the appropriate results for example:

10.1109/ICRoM.2018.8657641

10.1109/LARS/SBR/WRE54079.2021.9605425

Author Response

Dear reviewer, thank you for your feedback. I have tried to take into account all the following points

1- the decision-making process should be considered and defined by comparing the results.

To this end, the publication also includes a comparison with alternative methods. Likewise, the most important decisions and reasons for implementation are summarized in detail.

2- the analytical analysis should be added to the abstract and conclusion sections

A separate section has been included for the analysis of the problem.

3- the case study with precise results for a real-world system should be selected.

For this purpose, an additional example of an IRobot Roomba has been added to show its suitability in the real world. The full data can be found at the link below. To view the 3D visualization directly, the installation of a Python environment is required. (Only the data modules are available, which should be hopefully enough as evidence)

Additionally, a separate section was created on the details of the implementation. Communication is done via a REST API with Java script and a REST API for the digital twin. However, since the noise of the maps as well as the registration of multiple maps will be addressed in future work, only the update of the robot position in the map as well as the control of the robot itself will be presented here. Disadvantages and assumptions of the method as well as the implementation were added in the discussion.

4- the refs should be covered the all planning process in the introduction sections

The references as well as still others were also included. A use case was used to compare different alternative pathfinding methods with the presented method.

Shared Data and Source-Code for Path Planing:

https://faubox.rrze.uni-erlangen.de/getlink/fi867mQENWZXtZKR1jm2Z5/

I hope that I could address most of the critic.

Best Regrads

The author

Reviewer 2 Report

The work is on creating digital twins of dynamic robot paths and providing a framework for path planning based on density/traffic conditions. It's a nice and interesting topic for infrastructures. Congratulations to the authors. I think my comments below will contribute to the paper:

1- Scientific contribution and innovation are not emphasized enough. Please list items.

2- The study was carried out completely in computer environment. Simulation, code and/or data sharing should be done at least to editors and reviewers. Otherwise, suspicion may arise due to the possibility of plagiarism or fabrication and is a reason for rejection. Please provide relevant evidentiary content.

3- How will the proposed method be implemented in real application? The process of creating digital twins is well explained, but there is no clear explanation on how to ensure continuity. How will updating Twin data be carried out? What are the assumptions, limitations and deficiencies in the implementation?

4- The figures are beautifully colored. However, different color codes contain different meanings or states. It is very important to understand them when looking at a single picture. Please add each color/shape code as a legend to the figures. Figures are also used to summarize the process flow. In some cases it is difficult to find corresponding relationships. 

5- It is important to present the results of the study in a way that supports scientific contribution and to make comparisons/discussion with current studies.

Author Response

Dear reviewer,

thank you for your feedback. I have tried to take into account all the following points

1- Scientific contribution and innovation are not emphasized enough. Please list items.

The scientific contribution was inserted in the state of the art. Likewise, a comparison with alternative pathfinding methods can be found in the results as well as in the discussion.

2- The study was carried out completely in computer environment. Simulation, code and/or data sharing should be done at least to editors and reviewers. Otherwise, suspicion may arise due to the possibility of plagiarism or fabrication and is a reason for rejection. Please provide relevant evidentiary content.

For this purpose, an additional example of an IRobot Roomba has been added to show its suitability in the real world. The full data can be found at the link below. To view the 3D visualization directly, the installation of a Python environment is required. (Only the data modules are available, which should be hopefully enough as evidence)

3- How will the proposed method be implemented in real application? The process of creating digital twins is well explained, but there is no clear explanation on how to ensure continuity. How will updating Twin data be carried out? What are the assumptions, limitations and deficiencies in the implementation?

For this purpose, a separate section was created on the details of the implementation. Communication is done via a REST API with Java script and a REST API for the digital twin. However, since the noise of the maps as well as the registration of multiple maps will be addressed in future work, only the update of the robot position in the map as well as the control of the robot itself will be presented here. Disadvantages and assumptions of the method as well as the implementation were added in the discussion.

4- The figures are beautifully colored. However, different color codes contain different meanings or states. It is very important to understand them when looking at a single picture. Please add each color/shape code as a legend to the figures. Figures are also used to summarize the process flow. In some cases it is difficult to find corresponding relationships. 

As suggested, a legend with the RGB coding has been included.

5- It is important to present the results of the study in a way that supports scientific contribution and to make comparisons/discussion with current studies.

For comparison, an RGB map from a recent publication is shown to compare path finding with alternative methods.

Shared Data and Source-Code for Path Planing:

https://faubox.rrze.uni-erlangen.de/getlink/fi867mQENWZXtZKR1jm2Z5/

I hope that I could address most of the critic. Thank you for the detailed review.

Best Regrads

The author

Round 2

Reviewer 1 Report

Comments made in accordance with the recommendations.

Reviewer 2 Report

The improvements are enough. The paper can be accepted in present form.